# Analysing the Cost of Concentrated Feed and Income from Meat in Relation to Relative Growth Rate and Kleiber's Ratio in Intensive Fattening of Latvian Dark-Headed Lambs

**Ilva Trapina** [1,*] **, Daina Kairisa** [2] **and Natalia Paramonova** [1]

1 Genomics and Bioinformatics, Institute of Biology, University of Latvia, Jelgava str. 3, LV-1004 Riga, Latvia; natalia.paramonova@lu.lv

2 Institute of Agrobiotechnology, Faculty of Agriculture, Latvian University of Life Sciences and Technologies, Liela Street 2, LV-3001 Jelgava, Latvia; daina.kairisa@llu.lv

* Correspondence: ilva.trapina@lu.lv; Tel.: +371-29354786

**Abstract:** One of the goals of the breeding program of the Latvian national sheep breed Latvian dark-head (LT; Latvijas tumšgalve) is the improvement of meat productivity, in parallel with good reproduction characteristics. The main part of the costs is the cost of feeding, which can be reduced by raising sheep with high feed efficiency. This goal can be achieved by selecting sheep based on feed efficiency indicators. The fattening data of 100 LT lambs were analysed using the relative growth rate (RGR) and Kleiber's ratio (KR) indicators. The statistical difference was determined between low- and high-efficiency groups by calculating the cost of concentrate feed and income from meat realisation. The difference in the cost of concentrate feed was found after 60-days of intensive fattening and on the day needed for fattening up to 50 kg of live weight of lambs. When calculating the income of lamb meat, the differences between low and high RGR and/or KR efficiency groups were established. Thus, selecting high-ranking RGR and KR sheep can improve the cost and/or income of LT sheep breeders in Latvia.

**Keywords:** feed efficiency; feed cost; Latvian national breed; meat relation; lamb breeding

## 1. Introduction

Latvian dark-head (LT; Latvijas tumšgalve) is the only sheep breed of local origin selected in Latvia. The first breeding sheep herd book in Latvia was published in 1939. In 2014, eight breeding lines were reconfirmed in the population of the Latvian blackhead breed [1]. Currently, the live weight of full-grown ewes of the Latvian dark-head breed is about 55–65 kg, and rams are 95–120 kg. The average fertility of ewes is 150–160% [2].

The goal of LT breed selection for sheep breeders is to improve the breed's meat productivity, as LT is based on a mother breed with high reproducibility. To achieve this goal, LT has been crossed with other breeds for which a specific trait is more developed [3]. LT breed rams have been analysed using lamb fattening indicators in recent years [4,5].

According to the EUROSTAT, sheep and goat meat production in Latvia was 0.47 thousand tonnes in 2022, but in 2021, it was 0.45 thousand tonnes [6]. This year's average sheep meat price is EUR 395.41 for 2021 and EUR 475.93 for 2022 per 100 kg [7].

To make a profit, the total costs of breeding must be less than the income from the sale of meat. From total costs, variable costs represented 64.15%, operational fixed costs represented 21.66%, and 14.19% by the income of the factors (interests on fixed and working capital). Variable costs include all components involved in the activity that only occur if there is production and are directly related to the number of animals produced. The components are feeding (63.17%) and veterinary (0.98%) expenses [8]. Improving animal performance should be sustainable and economically profitable; therefore, economic selection indices have been used in most breeding programs for livestock [9]. The potential

profit from the sale of the animal depends on the timing of fattening or rearing the lambs, but this can become an economically unprofitable practice due to the cost of feeding [10]. So, the main cost item is feeding, which can be reduced by raising sheep with a high feed efficiency.

Feed efficiency indicators can be loosely described as ratio traits: feed efficiency (FE), feed conversion ratio (FCR), relative growth rate (RGR), and Kleiber's ratio (KR), or residual traits: residual feed intake (RFI), residual weight gain (RWG), and residual intake and body weight gain (RIG) [10,11]. Feed efficiency is the main indicator of fattening performance in farming production. With meat prices at their highest, not always compensating for increases in feed prices, the feed conversion ratio becomes the first economic indicator to improve [12].

In order to determine five of the seven parameters, it is necessary to know the exact average amount of food taken by the animal per day. However, in the case of RGR and KR, only the parameters of the animal's weight are used in the calculation. The Kleiber's ratio, the ratio of growth rate to metabolic weight, has been proposed as a valuable indication of growth efficiency and an indirect criterion for selecting feed conversion [13]. However, RGR is the accurate indicator of growth rate, representing the percentage increase in weight over a specific period. It dismisses the biological and physical measurement of time and can be directly employed to compare the development of individuals with significantly different weights or even across different species [14].

Previous studies have analysed the intensive fattening indicators [3,5,15]; the novelty of this study lies in the comparison of economic benefits—price for combined, concentrated feed and price for meat, in groups of different levels of feed efficiency indicators (relative growth rate and Kleiber's ratio), to identify indicators that could improve the expenses and/or income of sheep farmers. The results of such an analysis have a twofold benefit. First, the effect of feed efficiency on the economic situation in sheep farming is scientifically proven and can also be transferred to other areas of agricultural animals. Second, sheep farmers are given a tool to build their flock for a more economical result. In Latvian sheep farming, there are few scientific studies on the impact of feed efficiency on profitability.

## 2. Materials and Methods

### 2.1. Animals of Intensive Fattening

Based on the requirements of the breeding program of the Latvian dark-headed breed [2], every year, the offspring of certified sire rams are selected for breeding to find out the qualification of the sire ram to fattening indicators. At the ram breeding control station "Klimpas" in cooperation with the association Latvian Sheep Breeders Association in the summer months of 2022 and 2023, 100 ram lambs were fattened from 26 purebred Latvian dark-headed rams (three (3) or four (4) lambs per sire ram).

The age of the animals at the beginning of the control fattening was $82.75 \pm 7.39$ days with an interval from 66 to 106 days (2.2–3.5 months), and the initial body weight was $23.12 \pm 2.38$ kg. Before the start of intensive fattening and at the end of the process, the lambs were weighed on electronic scales with an accuracy of 0.01 kg.

According to the intensive fattening control protocol, all progeny from one sire ram were fattened in one pen of approximately 4 m², equipped with a loose tank for combined, concentrated feed and a slatted tank for hay. Straw was used as bedding. The shelter had natural ventilation through slits in the ceiling and windows equipped with anti-insect nets.

After a 7- to 14-day adaptation period, intensive fattening was initiated in the lambs. During the study, lambs were fed unlimited amounts of the combined, concentrated feed ($893.0$ g kg$^{-1}$ of consumed dry matter with $181.1$ g kg$^{-1}$ crude protein and $12.69$ MJ kg$^{-1}$ metabolizable energy) and hay [16]; in addition, mineral feed and licks salt were ensured. Water was provided from automatic waterers without limit.

The use of hay in the feeding process has been previously studied, and it has been determined that approximately 20% is eaten, and the rest is dropped to the ground during consumption [16]. Therefore, it is very difficult to determine the exact amount of hay eaten.

The keeping of animals during the research met the animal welfare requirements.

### 2.2. Fattening Variables

Based on changes in body weight adjusted for the 90th (BW90) and 150th (BW150) lamb days during 60 days of intensive fattening, the average daily gain (ADG) for each lamb was calculated. In a previous study [5], it was determined that there is no statistically significant difference between the initial body weight of the lambs, including the calculated body weight on the 90th day of life for LT lambs. In addition, AGD was calculated for all lambs of one pen together to find out the total AGD of one pen (APG). This variable was used to calculate the feed efficiency of the pen or all lambs of one ram.

The dry matter intake (DMI) was determined for concentrated feed only for one pen or ram lamb for all fattening periods (pDMI) and one feed day per lamb as the average DMI for the pen.

The slaughter weight was calculated by considering the percentage of the meat obtained at slaughter from the live weight of each lamb at the time of processing. The percentage of slaughter weight was used to determine the lamb meat mass (kg) at the 150th day of life and the lamb meat mass in kilograms obtained during the 60-day feeding period.

### 2.3. Feed Efficiency Variables

The obtained ADG and DMI of one pen, as well as live weight values, were used to calculate the feed conversion ratio (pFCR) and relative growth rate (pRGR) for each pen, and the ADG for each lamb was used to calculate the relative growth rate and Kleiber's ratio for lambs by using formulas previously published [10,11]: relative growth rate, % of BW per day, for a 60-day period is calculated as $(\log BW90 - \log BW150)/60$; Kleiber's ratio is $100 \times ADG/BW^{0.75}$, where $BW^{0.75}$ is metabolic BW from final BW (BW150); feed conversion ratio is DMI/ADG.

### 2.4. Dry Matter Intake or Concentration Feed Intake for Lamb

To analyse the cost of intake feed, an approximate DMI (aDMI) was calculated for each lamb. The DMI was calculated using the RGR (pRGR) and FCR (pFCR) of each pen and the RGR (lRGR) of each lamb. First, the ratio of pen RGR to each lamb's RGR (pRGR/RGR) was determined, thus identifying which lamb in the pen or for one ram had a higher and which had a lower RGR, compared to the pen's RGR or the average RGR among the lambs. Determining that there is a strong correlation between the pRGR and pFCR scores of the pen or ram [5], it is assumed that there is also a correlation between the RGR and FCR of each lamb, which is also consistent with the results from our previous [15] and other studies [13,14,17,18]. The thus obtained ratio between the RGR indicators is used to determine the individual FCR of each lamb from the pen's FCR or pFCR. The result is information on which lambs are above the average pFCR and below. Using the calculated FCR of the lamb, an approximate DMI was calculated using the formula for FCR and the ADG for each lamb.

The resulting aDMI is not considered 100% accurate for each lamb's DMI but is used to determine feed cost differences. The total sum of the individual aDMI of the lambs in one pen coincides with the total DMI of the pen, which is calculated for each pen. In addition, the average aDMI of the lambs in one pen corresponds to the DMI of one feed day per lamb or the average DMI of the pen.

### 2.5. Economic Data

Economic data for each lamb were calculated for the 60-day intensive feeding period and the 150th day so that the data would be mutually comparable.

In addition, the amount of concentrated feed needed for the lambs to reach up to 50 kg was calculated, along with the recommended live weight LT for the lambs at the time of slaughtering, based on their weight from the weight of the 70th day (lamb evaluation day

after weaning); information is according to the Latvian Breed Fattening Control Protocol (not published).

As a result of market research, it was found that the average cost of industrially prepared intensive fattening concentrated feed used in the study is EUR 13.65 per 20 kg (EUR 0.68 per kg). This price is specified for a concentrated feed for intensive fattening, manufactured industrially with minimal deviations from the standard composition. In certain farms with experience in feed mix preparation, owners use self-prepared concentrated feeds [19], significantly reducing their cost by EUR 234 per ton (EUR 0.23 per kg). The above-concentrated feed prices were used in the calculations as "feed cost_1" and "feed cost_2", respectively.

The price of mutton at the time of calculation, i.e., in February 2023, in Latvia, was EUR 564.7 per 100 kg [20] or EUR 5.65 per kilogram of meat. This amount will be used to calculate the economic benefit of the sale of meat.

Based on the feed cost for 60 days, the cost of concentrated feed per kilogram of live weight for this period was also calculated (feed cost per kg = feed cost/BW) to analyse the lamb cost.

Income over feed price (IOFC) for each lamb was calculated for 60 days as the difference between the concentrated feed cost of the intensive feeding period and the income, or income from the 60 days of meat gain: IOFC = feed cost − income from meat.

In addition, the ratio of feed costs to income from meat, feed cost/income from meat × 100%, was determined to analyse how much of the meat income is the cost of the concentrated feed. Calculations were made for a 60-day feeding period.

### 2.6. Statistical Analyses

General and analytical statistics were performed with SPSS v.25 (IBM Corp., NY, USA, 2017). The mean and standard distribution (SD) were calculated for measurement data, and lamb data were divided (Table 1) and analysed according to each indicator in three groups: low (<mean − 0.5 SD), medium (mean ± 0.5 SD), and high (>mean + 0.5 SD). Only the lowest and highest score groups were used to analyse variance to show the significance of the RGR and KR scores.

To calculate the difference between the RGR or KR groups of animals, an ANOVA was used in the presence of a normal distribution and the Kruskal–Wallis test in the absence of a normal distribution in at least one group, and a post-hoc test between groups. A significant result was determined if $p < 0.05$.

## 3. Results

### 3.1. Lamb's Groups

The average daily live weight gain for all 100 LT lambs was 0.33 ± 0.06 kg, which resulted in an average gain of 20.00 ± 3.70 kg over 60 days (from the 90th to 150th day of life) or body weight at the 150th day of 45.60 ± 5.09 kg.

When considering the RGR indicator, the difference between the groups of lambs with high and low values averaged 0.22% of the final body weight per day (Table 1). The difference in the Kleiber ratio indicator among groups of lambs was more than 1.35 times between the low and high groups of KR.

**Table 1.** The lambs were divided into three groups according to the relative growth rate (average for all lambs 0.36 ± 0.09) and Kleiber's ratio (18.93 ± 2.58) indicators.

| Indicator | Low | | Medium | | High | |
|---|---|---|---|---|---|---|
| | Value Border | Lambs (n; %) | Value Border | Lambs (n; %) | Value Border | Lambs (n; %) |
| Relative growth rate | <0.31 | 35    35.00 | 0.31–0.41 | 42    42.00 | >0.41 | 23    23.00 |
| Average ± SD | | 0.27 ± 0.03 | | 0.36 ± 0.03 | | 0.49 ± 0.06 |
| Kleiber's ratio | <17.65 | 31    31.00 | 17.65–20.21 | 35    35.00 | >20.21 | 34    34.00 |
| Average ± SD | | 15.99 ± 1.40 | | 18.92 ± 0.80 | | 21.62 ± 1.38 |

n—sample number in group.

### 3.2. Body Weight Gain

The higher difference between groups in the analysis of average weight gain in 60 days was found when calculating the RGR parameter (Table 2). We determined that the average weight gain over the 60-day fattening period was 16.46 ± 2.13 kg (ADG: 0.27 ± 0.04 kg) in a low-RGR group of lams and 24.13 ± 2.66 kg (ADG: 0.40 ± 0.04 kg) in high-RGR animals; the difference in weight gain between these groups of lambs at day 150 was 7.67 kg.

**Table 2.** Differences in feed efficiency indicators when analysing live weight indicators of lambs.

| | Body Weight | Value (Mean ± SD) in a Group of RGR or KR | | | |
|---|---|---|---|---|---|
| | | Low | Medium | High | \|H-L\| |
| Relative growth rate | ADG, kg | 0.27 ± 0.04 [a;b] | 0.34 ± 0.04 [a;c] | 0.40 ± 0.04 [b;c] | 0.13 |
| | On 150th day, kg | 42.38 ± 3.55 [a;b] | 46.91 ± 4.11 [a] | 48.11 ± 6.26 [b] | 5.73 |
| | In 60-day period, kg | 16.46 ± 2.13 [a;b] | 20.68 ± 2.18 [a;c] | 24.13 ± 2.66 [b;c] | 7.67 |
| Kleiber's ratio | ADG, g | 0.27 ± 0.03 [a;b] | 0.33 ± 0.03 [a;c] | 0.40 ± 0.04 [b;c] | 0.13 |
| | On 150th day, kg | 43.04 ± 3.7 [b] | 45.32 ± 4.73 [c] | 48.22 ± 5.37 [b;c] | 5.18 |
| | In 60-day period | 16.15 ± 2.06 [a;b] | 19.82 ± 1.89 [a;c] | 23.69 ± 2.35 [b;c] | 7.54 |

RGR—Relative growth rate; KR—Kleiber's ratio; SD—standard deviation; H-L—the difference between high- and low-efficiency groups; [a,b or c]—statistical significance ($p < 0.01$) between groups with the same letter within the indicator with post-hoc test.

### 3.3. Concentrated Feed Cost in Fattening Time

The difference in the average consumption of concentrated feed throughout 60 days in the groups of high- and low-efficient lambs was found in the case of both indicators (Figure 1). According to the results, the approximate feed consumed in the cases of both indicators is higher for animals with higher efficiency. The difference in the case of RGR (Figure 1A) is 9.94 kg in 60 days, and in KR (Figure 1B)—slightly less, or 7.32 kg. Accordingly, this means that when animals are divided according to RGR, low-efficiency animals cost EUR 6.76 (feed cost 1) or EUR 2.34 (feed cost 2) less over 60 days, and in the case of KR—by EUR 4.98 (feed cost 1) or EUR 1.72 (feed cost 2) less.

It is assumed that the price of concentrated feed per kilogram of live weight (feed cost per BWkg) for the most economical lamb should be as low as possible. The results show that in both RGR and KR, animals with higher efficiency consume more concentrated feed, but their ADG is also higher. Accordingly, it is important to determine whether more feed consumed is more economically beneficial per kilogram of ADG or BW obtained. The biggest difference in feed cost per kilogram of body weight between the low- and high-efficiency groups of lambs was found when calculating the Kleiber's ratio, when the average cost of concentrated feed per kilogram of body weight in animals of the low group was EUR 3.84 ± 0.49 or EUR 1.30 ± 017, respectively, by using market feed cost or self-made feed cost, and in the high group—EUR 2.84 ± 0.28 or EUR 0.96 ± 0.09, making a difference between groups of EUR 1.01 or EUR 0.34 per kg of body weight per animal.

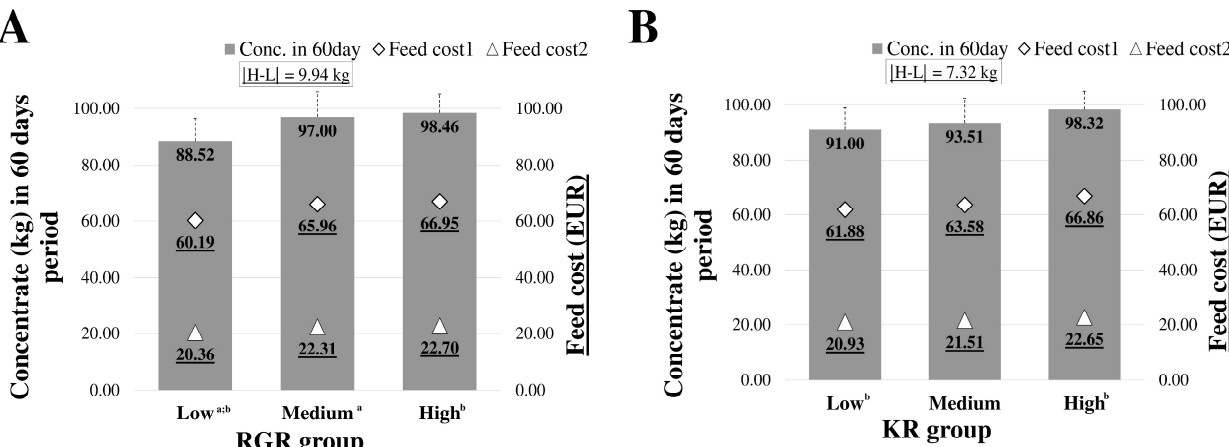

**Figure 1.** Differences in low, medium, and high relative growth rate (RGR; (**A**)) and Kleiber's ratio (KR; (**B**)) groups of lambs compared by weight, kg, and cost of concentrated feed, EUR, over a 60-days period. Feed cost 1—market price, EUR 0.68 per kg; Feed cost 2—self-made feed price, EUR 0.234 per kg. H-L—the difference between high and low-efficiency groups; [a,b or c]—statistical significance ($p < 0.01$) between groups with the same letter within the weight of concentrated feed with post-hoc test.

Considering certain differences between high- and low-performing groups of lambs for feed consumed during intensive fattening in 60 days, we determined this estimated difference in the period before slaughter. According to the Latvian Breed Fattening Control Protocol (unpublished), lambs from the same ram that are fed together are fattened to an average lamb weight of 45 to 50 kg. The feeding duration of up to 50 kg was calculated from the weight of the 70th (average $18.94 \pm 3.50$ kg), which is the average day when lambs are evaluated after weaning. Accordingly, the result is an approximate time for the lamb to be fully fattened.

The higher difference between groups was found when calculating the KR parameter (Figure 2B). Low-KR lambs need an average of a 27-day-longer intensive fattening period to reach a 50 kg live weight from the weight of the 70th day than high-KR lambs. For lambs of the low-KR group, the average weight of the 70th was $16.64 \pm 2.69$ kg, but for high-KR, it was $21.51 \pm 2.32$ kg.

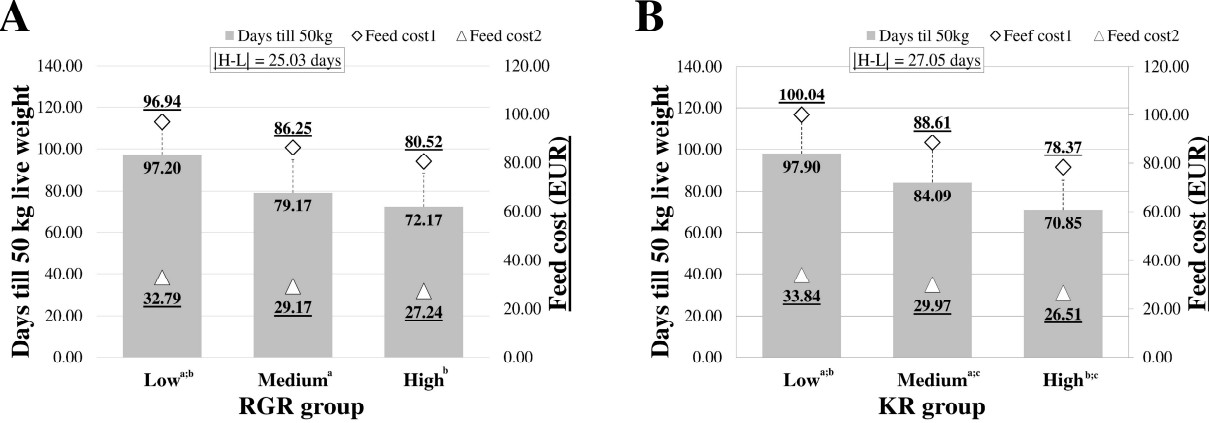

**Figure 2.** Differences in groups of lambs with low, medium, and high relative growth rate (RGR; (**A**)) and Kleiber's ratio (KR; (**B**)) groups compared by the intensive fattening period, days, to reach 50 kg body weight and the cost of concentrated feed, EUR, for this period. Feed cost 1—market price, EUR 0.68 per kg; feed cost 2—self-made feed price, EUR 0.234 per kg. H-L—the difference between high- and low-efficiency groups; [a,b or c]—statistical significance ($p < 0.01$) between groups with the same letter within the intensive fattening period with post-hoc test.

The final variable feeding costs are EUR 21.67 per animal (EUR 2167 per 100 animals) with feed cost_1 or EUR 7.37 per animal (EUR 733 per 100 animals) with feed cost_2.

### 3.4. Income from Meat Realisations

The marketed part of the animals is the slaughter weight, which in the case of LT lambs is, on average, 43.62 ± 1.99% of the live weight at the time of slaughter. The slaughter weight percentage from live weight does not statistically reliably differ between low- and high-efficiency lambs for feed efficiency indicators (Table 3).

**Table 3.** Differences between groups of feed efficiency indicators when analysing lambs' slaughter or carcass weight and income from it.

| | Slaughter or Carcass Weight | Value (Mean ± SD) in a Group of RGR or KR | | | | Income from Meat, EUR (Mean ± SD) per One Lamb in a Group of RGR or KR | | | |
|---|---|---|---|---|---|---|---|---|---|
| | | **Low** | **Medium** | **High** | **\|H-L\|** | **Low** | **Medium** | **High** | **\|H-L\|** |
| Relative growth rate | % from BW | 43.44 ± 2.16 | 43.82 ± 2.16 | 43.54 ± 1.33 | 0.10 | - | - | - | - |
| | weight, kg, from BW150th | 18.43 ± 1.92 [a;b] | 20.56 ± 2.12 [a] | 20.95 ± 2.80 [b] | 2.52 | 104.11 ± 10.86 [A;B] | 116.16 ± 12.00 [A] | 118.34 ± 15.83 [B] | 14.23 |
| | weight, kg, from 60-day gain | 7.15 ± 0.97 [a;b] | 9.06 ± 1.04 [a;c] | 10.50 ± 1.13 [b;c] | 3.35 | 40.39 ± 5.46 [A;B] | 51.20 ± 5.88 [A;C] | 59.32 ± 6.41 [B;C] | 18.93 |
| Kleiber's ratio | % from BW | 43.78 ± 2.47 | 43.67 ± 1.91 | 43.43 ± 1.59 | 0.34 | - | - | - | - |
| | weight, kg, from BW150th | 18.87 ± 2.19 [b] | 19.82 ± 2.46 | 20.93 ± 2.36 [b] | 2.07 | 106.59 ± 12.38 [B] | 111.98 ± 13.90 | 118.27 ± 13.31 [B] | 11.68 |
| | weight, kg, from 60 day gain | 7.08 ± 1.05 [a;b] | 8.67 ± 1.01 [a;c] | 10.28 ± 1.02 [b;c] | 3.21 | 39.98 ± 5.95 [A;B] | 48.98 ± 5.67 [A;C] | 58.09 ± 5.75 [B;C] | 18.11 |

RGR—relative growth rate; KR—Kleiber's ratio; SD—standard deviation; H-L—the difference in cost between high- and low-efficiency groups; [a, b or c/A, B or C]—statistical significance ($p < 0.01$) between groups with the same letter within the indicator with post-hoc test.

The study analysed the difference in the weight of the lambs on the 150th day of life and the growth for 60 days (Table 3). On average, the meat income of one animal for a lamb of the LT breed after 60 days in the feeding period is EUR 49.28 ± 9.30 at a purchase price of EUR 5.65 per kg.

The biggest difference in meat income from one animal was found between low-RGR and high-RGR groups of lambs. Slaughter or carcass weight over a 60-day feeding period was 7.15 ± 0.97 kg in low-RGR and 10.50 ± 1.13 kg in high-RGR animals, with a total income difference of EUR 18.93 per animal (Table 3).

### 3.5. Feeding Expenses and Meat Realisation Income Ratio

The meat gain from the total live weight of the lamb is less than half, so the real cost of lamb calculated against the cost of concentrated feed is greater than the cost of feed against lamb body weight kilogram. The difference between the expenses of the 60-day intensive feeding period, or the cost of the concentrated feed, and the income from the meat calculated as the income over the concentrated feed price (IOFC) was used to determine the economic profitability of the lamb.

On average, the IOFC for the low- and high-KR group of lambs was negative if the feed cost was calculated with the market feed price. The difference between low-KR (EUR −21.46 ± 5.95) and high-FCR lambs (EUR −8.68 ± 5.63) was on average, EUR 14.54 (Figure 3) per animal (EUR 1454 per 100 animals) in a 60-day fattening period if the industrial concentrated feed is used.

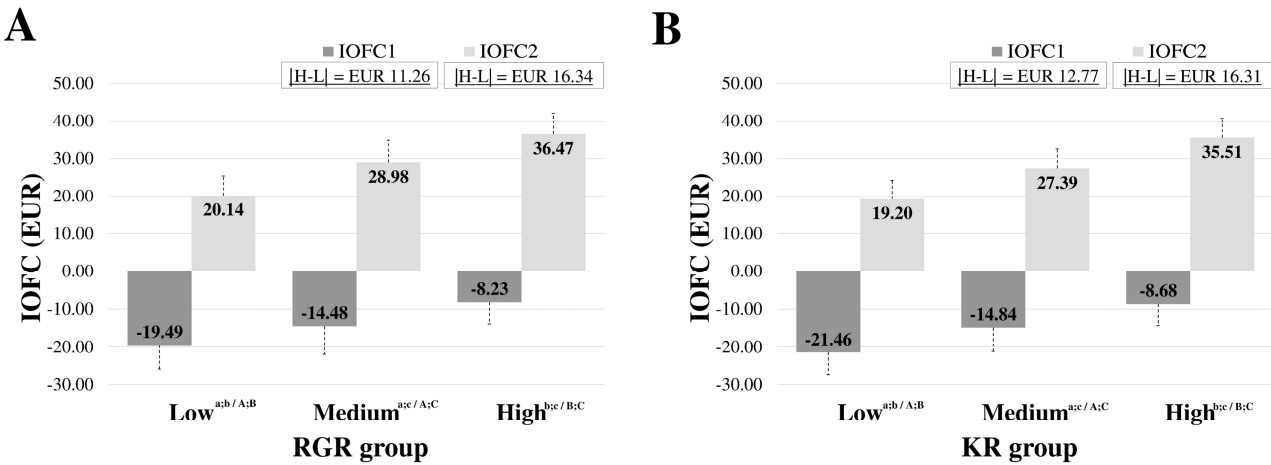

**Figure 3.** Differences in low, medium, and high relative growth rate (RGR; (**A**)) and Kleiber's ratio (KR; (**B**)) groups compared by income over feed price (IOFC), EUR, for a 60-day intensive fattening period. Feed cost 1—market price, EUR 0.68 per kg; feed cost 2—self-made feed price, EUR 0.234 per kg. H-L—the difference in cost between high- and low-efficiency groups; [a, b or c/A, B or C]—statistical significance (*p* < 0.01) between groups with the same symbol within the IOFC with post-hoc test.

In the case of self-made concentrated feed, the final difference is an average EUR 16.31 per animal (EUR 1631 per 100 animals) in a 60-day fattening period, but the IOFCs are positive in both groups. The income for the self-made concentrated feed price for the low-FCR group of lambs was 19.20 ± 5.03 EUR on average, but for high-FCR lambs, it was EUR 35.51 ± 5.25 (Figure 3), which amounted to a more than twofold difference.

The results show that in the case of low-KR, the concentrated feed cost is 155.75 ± 20.8% or 52.68 ± 7.04% of the meat realisation income obtained per animal. However, in the case of high-KR, the cost is 115.64 ± 10.51% or only 39.11 ± 3.56% of the meat realisation income per lamb (Figure 4).

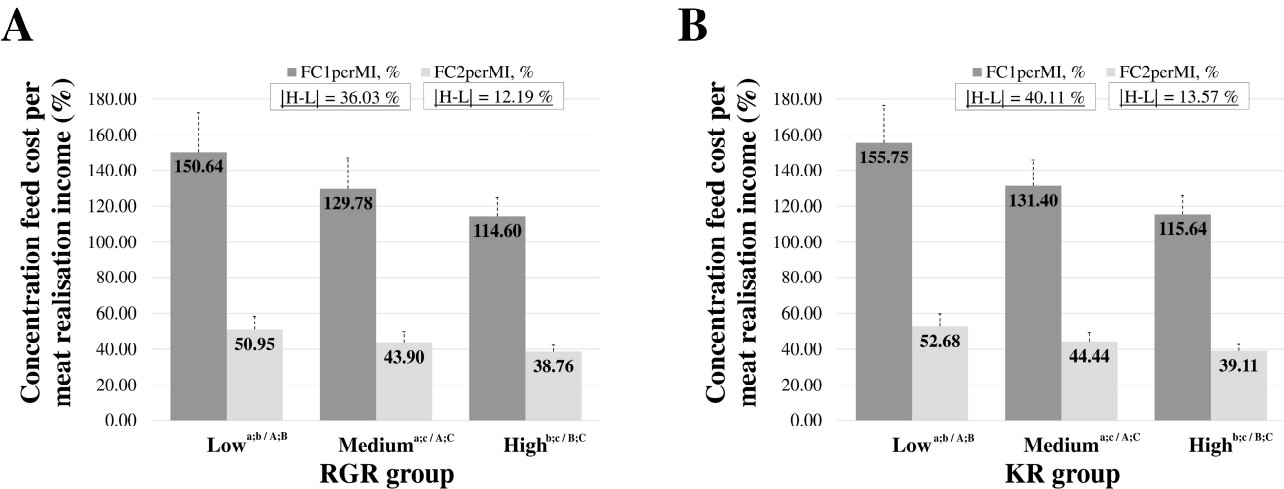

**Figure 4.** Differences in low, medium, and high relative growth rate (RGR; (**A**)) and Kleiber's ratio (KR; (**B**)) groups compared by concentrated feed cost per meat realisation income, %, for a 60-day intensive fattening period. Feed cost 1—market price, EUR 0.68 per kg; feed cost 2—self-made feed price, EUR 0.234 per kg. H-L—the difference in cost between high- and low-efficiency groups; [a, b or c/ A, B or C]—statistical significance (*p* < 0.01) between groups with the same symbol within the concentrated feed cost per meat realisation income, %, with post-hoc test.

## 4. Discussion

The Latvian dark-headed sheep is the only breed created by selection in Latvia. It is completely adapted to Latvia's climatic conditions and has the value of being a national breed [3]. Therefore, its preservation and improvement are not only the tasks of sheep breeders but they are also of national importance.

Collecting data on the intensive fattening of Latvian dark-headed sheep lambs and feed efficiency indicators and using these indicators in economic calculations allowed for an analysis of the possible economic significance of feed efficiency indicators in sheep meat breeding. The novelty of this research is in the comparison of economic benefits—the combined price of concentrated feed and the price of meat—in groups of different levels of feed efficiency indicators in order to identify feed efficiency indicators that could improve the expenses and/or income of the sheep farming farm. The results of such an analysis have a twofold benefit. First, the influence of feed efficiency on the economic situation in sheep farming has been scientifically proven and can be transferred to other areas of agricultural animals. Second, sheep farmers are given a tool to build their flock for a more economical result.

The Latvian Sheep Breeders Association's fattening of lambs is carried out as part of a breeding project, but it is completed for breeding purposes: to find out which ram has the offspring with the highest feed processing. Thus, the individual DMI of each lamb is not determined but for three or four lambs together it is. Accordingly, it was necessary to calculate the approximate DMI for each lamb using previously known correlations between feed efficiency indicators [13,14,17,18]. Therefore, we did not obtain 100% accurate data on the feeding of each lamb. In the following years, in cooperation with the Latvian Sheep Breeders Association, this could be changed by setting an automatic feeding system, with the help of which accurate feeding is recorded.

However, determining DMI is very important because this quantity affects ruminants' productivity and cost [21]. The study looked for an opportunity to calculate DMI. In sheep or small ruminant research, calculating DMI without direct counting is especially important when animals are fed grass/hay and legumes. With such feeding, DMI calculations can use internal markers determined in the feed and faeces [14,21].

However, this is not feasible when animals are kept and fed in groups—in a pen, such as in commercial farms. In cattle breeding, it is known that fattening animals in a group or one pen is more economical [22]. Over several years, various mathematical models have been developed in cattle breeding to determine an individual animal's intake/required feed from the total pen intake. The models are based on feed type, growth observations, BW, and carcass measurements [23,24]. Modifications and compilations of these models are included in the Cornell/Cattle Value Discovery System (CVDS) mathematical program. Currently, the modification of the CVDS model includes the dynamic, iterative, mechanistic (DIM) model to calculate the amount of feed needed for cattle to obtain a specific body structure (amount of fat, carcass quality) in a specific period. The model can predict ADG when DMI is known, or dry matter is required when ADG is known [22].

An adapted model for sheep, the Small Ruminant Nutrition System (SRNS; https://www.nutritionmodels.com/srns.html (accessed on 3 February 2024)), is also available [23], with the help of which it is possible to determine the nutrient requirements and biological values of feeds. However, it should be mentioned that this particular model is designed to determine how much feed is needed for different purposes, not how much is consumed. Using the previously mentioned model for sheep, it would be possible to determine the desired DMI, taking into account different parameters like breed, BW, feed, etc. [23], and compare it with the estimated approximate DMI, so it would also be possible to determine the RFI for each animal in future studies. However, we need to consider that this model has not been tested for the Latvian dark-head breed.

Considering that the DMI was not precisely determined, we did not use it in the calculations to determine feed efficiency indicators such as FCR, RFI, RWG, and RIG, but in the analysis of lambs, we used those feed efficiency indicators (RGR and KR) that only

include body weight and ADG [10,11]. However, when analysing the differences obtained in general, we consider the previously determined correlations between all indicators of feed efficiency [5,13–15,17,18], which allows for the conclusion of feed efficiency in general.

At the control feeding station, lambs are fed with the highest quality concentrated feed, purchased from the manufacturer at a high price (market price 0.68 per kg). However, in real conditions, farmers use mixtures of concentrated feed prepared from purchased raw materials on sheep farms. Under these conditions, the identical composition of the feed is not ensured, as a condition necessary for controlling fattening. However, the final price for feed produced by the sheep owners themselves is lower—EUR 0.234 per kg [19]. Thus, if the correct proportions of the necessary components of concentrated feed are observed, reducing the cost of feeding animals seems possible. Therefore, we used both costs in economic calculations, showing the possibility of fattening lambs in different cost categories while maintaining quality feeding.

Despite the importance of feed efficiency indicators in sheep breeding and the economy [10], only a few studies have compared groups of high- and low- efficiency sheep. In addition, it should be mentioned that in these studies, the animals are classified according to the RFI, the calculation of which requires an accurate determination of the DMI. For example, the average difference between low and high RFI values is about 160 g per day for Pelibeuy sheep, which received food with slightly less metabolizable energy [25], or around 380 g per day for ½ Dorper × ½ Santa Inês male lambs [10]. However, it should be mentioned that sheep breeders in Latvia who are involved in the selection of LT breeds do not have systems for automatically determining the amount of feed, and it is impossible to determine the exact amount eaten by each lamb. That means studies need to be conducted where RGR and/or KR determine feed efficiency.

According to published data, a 10% improvement in ADG due to a 7% increase in feed intake increases profitability by 18%, and a 10% improvement in feed efficiency leads to a 43% increase in profits. Thus, efforts to improve feed efficiency use can significantly reduce input costs for meat production [26]. Our study also aimed to determine whether such an effect of feed efficiency parameters is present in LT lambs.

Lambs in intensive fattening averaged 82.75 ± 7.39 days, with an interval from 66 to 106 days (2.2–3.5 months). Accordingly, three live weight values were calculated to mutually compare lambs: the 90th day of life as the starting point, the 150th day as the endpoint, and the 60-day feeding period [5]. The difference in weight gain between RGR groups of lambs on day 150 was 7.67 kg.

The difference in the average consumption of concentrated feed throughout 60 days in the groups of high- and low-efficient lambs was found. However, when dividing the animals into groups according to RGR or KR, the more efficient animals ate more. These survivors also ended up gaining more weight. Considering that we used the estimated DMI of each animal in the calculations, we could not calculate the RFI with sufficient reliability. It would help to understand whether the effectiveness of high RGR and KR in making animals eat more is also against the planned DMI or not [27].

According to the Latvian Breed Fattening Control Protocol (unpublished), lambs from the same ram fed together are fattened to an average lamb weight of 45 to 50 kg, which is also the fattening weight of lambs recommended by Latvian sheep breeders for the Latvian dark-headed sheep breed. When fattening above this weight, the amount of meat obtained in terms of days of fattening is significantly higher than for a weight of up to 50 kg [4]; also, the proportion of fat increases.

The feeding duration of up to 50 kg was calculated from the weight of the 70th day or average day when breeder experts evaluate lambs after weaning. The study found that high-KR lambs needed an average of 27 days less to reach a 50 kg live weight. Accordingly, the final variable feeding costs are EUR 21.67 per animal with feed cost_1 or EUR 7.33 per animal with feed cost_2.

The obtained results indicate not only that high-efficiency animals need to be fed for a shorter time but also that their starting weight around the 70th day is higher, which

indicates a better start for the animal. In our previous study [5], where only the data from 2022 were included, it was already established that even with similar birth weights, LT lambs reach a statistically different weight around 90 days. Accordingly, it is important to predict which ram will have high-feed-efficiency offspring in time.

Each fattening day is an additional cost that can only be compensated with income. It is assumed that the price of concentrated feed per kilogram of live weight (feed cost per BWkg) for the most economical lamb should be as low as possible. Thus, between two low- and high-KR herds of 100 animals each, with an increase in weight of 25 kg, the total difference in economic yield will be from EUR 849 by using self-made concentrated feed to EUR 2513 by industrial-made concentrated feed. Therefore, it is not economically viable to raise lambs with a low KR.

The marketed part of the animals is the slaughter weight, which in the case of LT lambs is, on average, 43.62 ± 1.99% of the live weight at the time of slaughter. Accordingly, sheep that have reached this weight are a profitable part of livestock. The slaughter weight percentage from live weight does not statistically reliably differ between low- and high-efficiency lambs for feed efficiency indicators. Accordingly, there is no statistically significant difference in slaughter weight when all lambs have the same live weight, for example, the recommended 50 kg. In real animal husbandry practice, slaughtering lambs is carried out depending on buyers' demand and to reduce transportation costs, considering the average weight of animals in a group. Therefore, the actual live weight of lambs at the time of slaughter can vary significantly.

To determine the economic profitability of the lamb, the income over the concentrated feed price (IOFC) was analysed. There is a need to consider that the IOFC is not a profit calculation, as the profit calculation must include other day-to-day expenses such as the price of hay, employee salaries, taxes, and transportation costs [8].

The results show that about an EUR 1454 IOFC is obtained in a breeding herd of 100 low- or high-feed-efficiency lambs, regardless of the feed price. However, obtaining a positive IOFC using a cheaper but equally high-quality feed is also possible. Not all sheep farmers have sufficient knowledge to prepare high-quality concentrated feed, so it is necessary to invest in sheep selection.

The results show that by increasing feed efficiency, it is possible to improve the IOFC even when using a more expensive concentrated feed. The difference between low-KR and high-KR lambs was EUR 12.77 on average per animal or EUR 16.31 on average per animal in a 60-day fattening period, depending on feed cost.

Currently, the IOFC has a high feed price of around EUR −15 in the medium feed efficiency group. However, through selection, the average indicator can be equivalent to the currently higher feed efficiency group indicator, or around EUR −8.6. Respectively, that would mean that the IOFC of the new high-feed efficiency group would be positive.

The overall analysis of the studied parameters of feeding efficiency in LT lambs, given in our article, shows that, on average, on the farm, for the LT breed, the cost of concentrated feed may be around 50% of the income to be received from the sale of meat. Including the other feed costs, the total feed cost can reach 60–70% of the income mentioned above [8,10], but it depends greatly on feed price, feed quality, and sheep breed. However, by improving the efficiency of feed processing, it is possible to reduce this percentage and increase income.

Our results demonstrate statistically significant differences between the low and high groups for both RGR and KR. The differences between RGR and KR data are not large in one or another indicator. Both indicators show the efficiency of the animal. Furthermore, taking into account the correlations of these indicators with other indicators of feed efficiency [5,13–15,17,18], we can conclude that each of the indicators of feed efficiency would allow for improving sheep breeds. The choice of which indicators to include in the breeding program depends on the ability to determine the exact DMI.

Based on the analysis's results, we can conclude that improving feed efficiency indicators in the breeding process is necessary, thus ensuring both breed improvements

and economic benefits for growers. However, sheep breeders' education is also of great importance, and it is available in universities and qualification courses.

## 5. Conclusions

Based on the results obtained from intensive fattening of the Latvian dark-headed breed, the economic benefit of the LT breed is profitable and similar to that of different sheep breeds for meat. Analysing the approximate DMI values, a greater difference was found when the lambs were divided into groups according to RGR, but when analysing the meat sales income, a greater difference was found according to KR. The income over concentrated feed prices for higher-KR lambs are three times higher than the low-KR group, indicating the potential to be used in breeding.

Thus, selecting LT sire ram lamb by RGR and/or KR parameters can improve LT sheep breeding and fattening efficiency and promote the development of the meat industry in Latvia. In addition to educating sheep breeders/farmers about the composition of concentrated feed and the possibility of preparing it themselves, the economic situation in breeding LT sheep for meat can be improved.

**Author Contributions:** Conceptualization, I.T., N.P. and D.K.; data curation, I.T. and D.K.; formal analysis, I.T.; funding acquisition, N.P. and D.K.; investigation, I.T. and N.P.; methodology, I.T.; validation, I.T., N.P. and D.K.; writing—original draft, I.T.; writing—review and editing, I.T., N.P. and D.K. All authors have read and agreed to the published version of the manuscript.

**Funding:** Supported by the Latvian Council of Science, Latvia, Project LZP-2021/1-0489, project: "Development of an innovative approach to identify biological determinants involved in the between-animal variation in feed efficiency in sheep farming".

**Institutional Review Board Statement:** The research did not require ethical approval, as the creation of animal sample collections and related ethics in Latvia are regulated by laws of the Republic of Latvia: "Animal Protection Law" and "Veterinary Medicine Law", and Cabinet of Ministers Regulation No. 5, "General Animal welfare requirements for livestock" and No. 52, "Rules for the protection of animals for scientific use". In addition, based on Directive 2010/63/EU of the European Parliament and of the Council of 22 September 2010 on protecting animals used for scientific purposes, the procedures for the selected animals can be considered light.

**Data Availability Statement:** Data on feed efficiency indicators of LT lambs in 2022 can be found in Trapina Ilva. (2023). Database of feed efficiency indicators of intensive fattening lambs of Latvian sheep breeds within the framework of the Latvian Council of Science LZP-2021/1-0489, project [Data set] Zenodo: https://doi.org/10.5281/zenodo.8143244. Data sets or economic calculations used and analysed during the current study are available from the corresponding author upon reasonable request.

**Acknowledgments:** We thank the employees of the Latvian Sheep Breeders Association and the sire-control fattening farm Klimpa for their valuable help in sheep breeding and data acquisition.

**Conflicts of Interest:** The authors declare no conflicts of interest.

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
