# Peer review of "Analysing the Cost of Concentrated Feed and Income from Meat in Relation to Relative Growth Rate and Kleiber’s Ratio in Intensive Fattening of Latvian Dark-Headed Lambs"

_agriculture, doi:10.3390/agriculture14040593_

Round 1
Reviewer 1 Report
Comments and Suggestions for Authors
When intensively fattening Latvian dark-headed lambs, analyzing the cost of concentrated feed and income from meat about feed efficiency parameters is crucial.
I suggest to better present Latvian dark-head by reformulating the writing in the summary "One of the objectives of breeding the dark-headed Latvian sheep (LT; Latvijas tumšgalve), the only one originating from Latvia, is to improve meat productivity" This work highlights the great existing diversity that probably allows genetic improvement for meat production.
Keywords can be improved by not using words that are already in the title "Keywords: Feed efficiency; Concentrated feed cost; Latvian dark-head; Meat price; Lamb fattening”, for example, sheep breed of local, qualification of the ram, fattening indicators.
Lines 157-159 - “Lamb data were analysed according to feed efficiency, creating three groups: low (< 157 mean ‐ 0.5 SD), medium (mean ± 0.5SD), and high (> mean + 0.5SD) parameters (Supplementary Table S2)” properly should stay in the "Materials and Methods”.
Supplementary tables should be cited in the "Materials and Methods" and not in the results, if any of the tables need to be cited in the results it is indicative that they should not be supplementary but included in the article.
In figures 1 and 2 insert the number of animals representing the means presented for each level (low, medium and high)
Line 174 - I suggest replacing "Living weight gain" with "Body weight gain" considering it is more appropriate.
I didn't understand why the authors reference "concentrated feed" in the title and throughout the text. I suggest that "concentrated feed" be removed and replaced by "intake feed", as the selection capacity of sheep and the variation between individuals is important to correctly reference what was carried out "intake feed".
Author Response
Response to Reviewer 1 Comments
Reviewer 1
First of all, the authors would like to thank Reviewer#1 for reading the manuscript and giving important comments and suggestions. We have made a major revision of the manuscript according to all reviewers comments and below there are our point-to-point replies, explanations, and indications on changes that were done in the revised version to Reviewer#1 comments,.
Reviewer#1 (in bold):
When intensively fattening Latvian dark-headed lambs, analyzing the cost of concentrated feed and income from meat about feed efficiency parameters is crucial.
I suggest to better present Latvian dark-head by reformulating the writing in the summary "One of the objectives of breeding the dark-headed Latvian sheep (LT; Latvijas tumšgalve), the only one originating from Latvia, is to improve meat productivity" This work highlights the great existing diversity that probably allows genetic improvement for meat production.
The specific text was changed to: "One of the goals of the breeding program of the Latvian national sheep breed Latvian dark-head (LT; Latvijas tumšgalve) is the improvement of meat productivity, in parallel with good reproduction characteristics.", thus showing more the importance of the particular breed.
Keywords can be improved by not using words that are already in the title "Keywords: Feed efficiency; Concentrated feed cost; Latvian dark-head; Meat price; Lamb fattening”, for example, sheep breed of local, qualification of the ram, fattening indicators.
We changed the keywords to: “Feed efficiency; feed cost; Latvian national breed; meat relation; lamb breeding”.
Lines 157-159 - “Lamb data were analysed according to feed efficiency, creating three groups: low (< 157 mean ‐ 0.5 SD), medium (mean ± 0.5SD), and high (> mean + 0.5SD) parameters (Supplementary Table S2)” properly should stay in the "Materials and Methods”.
Supplementary tables should be cited in the "Materials and Methods" and not in the results, if any of the tables need to be cited in the results it is indicative that they should not be supplementary but included in the article.
When modifying the article, we removed all the tables in the appendix because they were no longer needed. Moreover, we left the explanation about the division of groups only in the "Materials and methods" section.
In figures 1 and 2 insert the number of animals representing the means presented for each level (low, medium and high)
Both tables mentioned were replaced by tables to represent better the results obtained. Thus also showing the size of the group divisions.
Line 174 - I suggest replacing "Living weight gain" with "Body weight gain" considering it is more appropriate.
The text was corrected as indicated by the reviewer.
I didn't understand why the authors reference "concentrated feed" in the title and throughout the text. I suggest that "concentrated feed" be removed and replaced by "intake feed", as the selection capacity of sheep and the variation between individuals is important to correctly reference what was carried out "intake feed".
The term “concentrate feed” is used in the text because both in the control station and on the farms sheep are fed a compound feed: concentrate and hay. Hay is often prepared on-site at the farms and the price is not fixed. In the control fattening station (1) hay is prepared on site during the summer period and its cost is not calculated and (2) there is a record of how much hay is used for one pen of lambs, but in previous studies, it has been found that the lambs in the control fattening station eat only 20 % of hay. As well as taking into account the recommendations of other reviewers, completely accurate determination of the amount of hay eaten is not possible.
According to our calculations with taking account previous studies, the "intake feed" or total feed intake on average consists of 7.76% hay and feed concentrate - 92.33%. But it is in average. Therefore, using the term "intake feed" would be incorrect only for "concentrated feed" calculations.
Reviewer 2 Report
Comments and Suggestions for Authors
In this type of study, all variables are measured as accurately as possible and this particular study shows a deficiency.
Firstly, feed and consequently dry matter intake must be measured individually even when the weight of the animals is apparently homogeneous. According to the methodology used, the classification of an animal in the low, medium or high group can be based on one hundredth of a unit. By dividing consumption among the group of animals, the opportunity to make a precise and objective classification is lost.
On the other hand, the presentation of the results should be improved, making them easier for the reader to understand. In their current presentation, it is very confusing to go from the supplementary material to the manuscript to understand it.
This study has valuable information that I suggest be presented as a case study or economic study and not as an experimental study.

Author Response
Response to Reviewer 2 Comments
Reviewer 2
First of all, the authors would like to thank Reviewer#2 for reading the manuscript and giving important comments and suggestions. We have made a major revision of the manuscript according to all reviewers comments and below there are in pdf file our point-to-point replies, explanations, and indications on changes that were done in the revised version to Reviewer#2 comments,.
Reviewer#2 (in bold):
In this type of study, all variables are measured as accurately as possible and this particular study shows a deficiency.
Firstly, feed and consequently dry matter intake must be measured individually even when the weight of the animals is apparently homogeneous. According to the methodology used, the classification of an animal in the low, medium or high group can be based on one hundredth of a unit. By dividing consumption among the group of animals, the opportunity to make a precise and objective classification is lost.
On the other hand, the presentation of the results should be improved, making them easier for the reader to understand. In their current presentation, it is very confusing to go from the supplementary material to the manuscript to understand it.
This study has valuable information that I suggest be presented as a case study or economic study and not as an experimental study.
Taking into account the recommendations of all reviewers, we made major changes to the structure of the article and data analysis. First, taking into account the possibility of making changes, we increased the number of samples to 100 (48 +52) lambs, which make up the 2022 and 2023 fattening groups. Second, we analyze only two measures of feed efficiency (Relative growth rate (RGR) and Kleiber’s ratio (KR)) because we do not have data available on exact individual DMI, but set this value as a calculated, approximate value to analyze the amount of concentrate feed eaten. Third, due to the reduced number of feed efficiency indicators, all Supplementary tables have been converted to figures or smaller tables inserted into the text.
Significant changes have been made to the manuscript of the article accordingly.
In addition to the comments and pointers that are in the pdf file, is answered in the pdf file by each comment.

Reviewer 3 Report
Comments and Suggestions for Authors
This manuscript describes an experiment evaluating economics of feed efficiency traits in Latvia sheep. There is a major concern with the experimental design that must be corrected before publication and may warrant rejection. The 48 lambs were fed in 13 sire group pens - individual feed intake of lambs was not measured. Therefore, feed efficiency traits CANNOT be calculated on individual lambs. The feed efficiency traits MUST be calculated on a sire group/pen basis; however, then there is only N = 13 observations which is not enough to capture much variation in feed efficiency. I suggest that the authors need to collect more data from the ram evaluation testing facility to increase the number of sire group/pen observations.
Further comments are in the attached document.

Comments on the Quality of English Language
English grammar is good for the most part. However, the word choices in many instances lead to a vague description of the results or meaning of the results. More specific language needs to be used - the words 'group', 'indicator', 'parameter', etc. are used without a describing adjective.
Author Response
Response to Reviewer 3 Comments
Reviewer 3
First of all, the authors would like to thank Reviewer#3 for reading the manuscript and giving important comments and suggestions. We have made a major revision of the manuscript according to all reviewers comments and below there are in pdf file our point-to-point replies, explanations, and indications on changes that were done in the revised version to Reviewer#3 comments,.
Reviewer#3 (in bold):
This manuscript describes an experiment evaluating economics of feed efficiency traits in Latvia sheep. There is a major concern with the experimental design that must be corrected before publication and may warrant rejection. The 48 lambs were fed in 13 sire group pens - individual feed intake of lambs was not measured. Therefore, feed efficiency traits CANNOT be calculated on individual lambs. The feed efficiency traits MUST be calculated on a sire group/pen basis; however, then there is only N = 13 observations which is not enough to capture much variation in feed efficiency. I suggest that the authors need to collect more data from the ram evaluation testing facility to increase the number of sire group/pen observations.
On behalf of the author's collective, I would like to express my gratitude for pointing out our mistake about using the average DMI. Lamb data were taken from sheep farmers who use control fattening to evaluate rams for selection programme not for science and use the average DMI there, as exact data is not needed, but data for agricultural evaluation only.
According to the directions, we changed the design of the study and revision of the manuscript.
- While waiting for the review, the final information about the fattened lambs of 2023 was received, so it was possible to increase the number of lambs to 100 (48 from 2022 and 52 from 2023), which is the offspring of 26 sire rams (or 26 pens).
- In the analysis for each lamb, using fattening data, we will use (in the revised manuscript) only two feed efficiency indicators: Relative growth rate (RGR) and Kleiber’s ratio (KR). Body weight and daily weight gain are used to calculate these indicators.
- To implement our idea of a difference in concentrated feed between high and low efficiency animals, we sought to calculate an approximate DMI for each lamb. For this, we use the information for total data of offspring of each sire ram (data from each pen) for which we determined the RGR and FCR, and based on this data we made calculations. In the revised manuscript, in the chapter "Materials and methods" was made section for explaining the calculation. Also only information on the consumption of concentrated feed is used in the calculations and analyses, because the information on the use of hay is very imprecise, and it is very difficult to determine the cost of hay.
Taking into account all the above, major revisions are made.
Further comments are in the attached document.
To the comments and pointers that are in the pdf file, is answered in the pdf file by each comment.
We did not respond to the comments that were written on the Supplementary tables in the PDF file, because the Supplementary tables are no longer needed in the revised manuscript. However, these comments will be taken into account.

Round 2
Reviewer 2 Report
Comments and Suggestions for Authors
The document improved substantially with the changes made, however it seems to me that the results tables can still be improved to make their interpretation easier.

Author Response
First of all, the authors would like to thank Reviewer#2 for reading the manuscript and giving important comments and suggestions. We have made a revision of the manuscript according to all reviewers comments and below there are in pdf file our point-to-point replies, explanations, and indications on changes that were done in the revised version to Reviewer#2 comments,.
Reviewer#2 (in bold):
The document improved substantially with the changes made, however it seems to me that the results tables can still be improved to make their interpretation easier.
We modified the tables to include the statistical analysis for all three groups and insert the results of the statistical post-hoc test.
We also answered the other comments in the pdf file.

Reviewer 3 Report
Comments and Suggestions for Authors
The authors have made improvements to the manuscript by removing the calculation of individual feed efficiency on pen level DMI. The calculation of DMI based on relationships between RGR and FCR has some errors and forces DMI to be related to ADG, which does not completely reliably capture the variation in DMI among individuals, but is an improvement. The authors should read and discuss the work by Luis Tedeschi's laboratory on predicting individual feed intake of group-fed animals.
Specific comments in the attached document.

Author Response
First of all, the authors would like to thank Reviewer#3 for reading the manuscript and giving important comments and suggestions. We have made a major revision of the manuscript according to all reviewers comments and below there are in pdf file our point-to-point replies, explanations, and indications on changes that were done in the revised version to Reviewer#3 comments,.
Reviewer#3 (in bold):
The authors have made improvements to the manuscript by removing the calculation of individual feed efficiency on pen level DMI. The calculation of DMI based on relationships between RGR and FCR has some errors and forces DMI to be related to ADG, which does not completely reliably capture the variation in DMI among individuals, but is an improvement. The authors should read and discuss the work by Luis Tedeschi's laboratory on predicting individual feed intake of group-fed animals.
Many thanks for the recommendations of L. O. Tedeschi's publications. These publications and the modified model for sheep will certainly be useful in future studies, however, so far these models have not been tested on the Latvian dark-headed breed. We very much hope that in cooperation with Latvian sheep breeders we will be able to use this model in the next year's control fattening process.
These models are designed to calculate and determine the amount of DMI required for a specific purpose, taking into account both the type of animal (sheep, lamb or ram; meat or dairy), feed components, housing conditions and other parameters, not to determine how much the lamb has eaten. This model (the equation in (Cannas et al., 2004; [23]), which also includes ADG) could be used in the future to calculate the required DMI and RFI for lambs.
We would also like to re-emphasize that the DMI calculation in this article is not considered to be the exact DMI for each lamb, but only an approximate DMI for economic calculations.
It should be mentioned that the amount of DMI will always be related to ADG, as the DMI required by a lamb depends on its weight and weight gain (DMI increases with increasing ADG). This is also indicated by the calculated DMI included in the RFI calculation, which is calculated using metabolic terminal weight and ADG.
In addition, taking into account the recommendation, we performed statistical analysis for all three groups. Until now, we analyzed only the extreme groups, to show the pronounced difference, why selection according to feed efficiency indicators is necessary in the future for the Latvian dark-headed breed, which until now has been selected more as a mother breed.
Responses to the comments in the pdf file are included in the attached pdf file.
